# Past Happiness and Broken Future Horizon of Oncological Patients during Chemotherapy—A Quantitative Exploration of a Phenomenological Hypothesis

**DOI:** 10.3390/cancers16112124

**Published:** 2024-06-02

**Authors:** Magdalena Fryze, Patrycja Wisniewska, Jadwiga Wiertlewska-Bielarz, Marcin Moskalewicz

**Affiliations:** 1Department of Psychology, Medical University of Lublin, 20-059 Lublin, Poland; 2Philosophy of Mental Health Unit, Department of Social Sciences and the Humanities, Poznan University of Medical Sciences, 61-701 Poznań, Poland; 3Phenomenological Psychopathology and Psychotherapy, Psychiatric Clinic, University of Heidelberg, 69115 Heidelberg, Germany; 4Institute of Philosophy, Maria Curie-Sklodowska University, 20-400 Lublin, Poland; 5IDEAS NCBR, Chmielna 69, 00-801 Warsaw, Poland

**Keywords:** lived experience, phenomenology, time, time perception, temporality, quality of life, qualitative research, chemotherapy

## Abstract

**Simple Summary:**

Cancer significantly affects physical and mental health, evoking intense emotions and forcing life changes. Emotional responses to cancer treatment have an intrinsic connection with time, in particular with the positive memories of how life looked before the disease and the often grim and hopeless outlook on the future. This study found that patients in chemotherapy experience increased focus on the present, decreased focus on the future, and a sense of unpredictability, with a relatively short temporal horizon measured in weeks and months, not years. These experiences are prevalent regardless of the type of cancer, the length of treatment, and the sociodemographic background. It can be concluded that changed temporal experience during chemotherapy is a factor that can have an impact on patients’ well-being and ability to cope with the disease.

**Abstract:**

Understanding the impact of cancer on the experience of time is crucial in the context of hope and recovery. This study, a follow-up to a previous qualitative study of ovarian cancer patients – explored two types of such experiences—the memory of past happiness and the limited future planning. A sociodemographic questionnaire with nine questions about the experience of time was used on a convenience sample of 202 patients with various cancers, predominantly women with breast, ovarian, and cervical cancer. It was found that the respondents experienced increased focus on the present, decreased focus on the future, and a sense of unpredictability, with a relatively short temporal horizon measured in weeks and months, not years. Almost half of the respondents (46%) measured time during treatment by the rhythm of chemotherapy and check-ups, which thus appeared as the most meaningful events. The increase in the frequency with which patients underwent chemotherapy mildly affected their focus on the present (R = 0.25, *p* < 0.05), likely because of the discomfort of the side effects. The correlations between age and time in treatment, on the one hand, and the experience of time, on the other, were negligible. Changed temporal experience during chemotherapy is a factor that can have an impact on patients’ well-being and ability to cope with the disease. It thus should be taken into account when planning oncology care.

## 1. Introduction

### 1.1. The Experience of Time in Cancer: Memories of the Past and Plans for the Future

Cancer is one of the most traumatic experiences that can befall a person. It affects physical and mental life, contributing significantly to the disruption of existence. From the moment of diagnosis to daily struggle with the disease, patients experience a wide range of emotions that affect their well-being, including sadness, anxiety, anger, frustration, and uncertainty about the future. Chronic oncological conditions disrupt the sense of time and reorganize its experience, also due to the rhythmic nature of treatment [1,2,3,4,5,6,7,8,9,10,11].

This study is concerned with understanding how cancer patients during chemotherapy experience time, especially regarding their recollections and plans. Time can be understood in various ways, either as clock time or a sequence of consecutive events or, from a phenomenological perspective, as a stream of experience permanently linked to our presence in the world [1,12,13]. From this perspective, memories and plans for the future are inseparable from present existence and affect the lived experience of reality. Also, they are not merely passive but are actively constructed by current perceptions, emotions, and the meaning attributed to them. Recollections and anticipations are intrinsic to lived presence, and their meaning changes depending on the context and individual development [14,15]. In the absence of positive or negative stimuli, there is usually no awareness of the passing of time, so it ceases to be explicitly felt, while a person experiencing positive emotions might have the sense of time accelerating or, when experiencing negative emotions or exposed to unpleasant events, the sense of time dragging. A disturbed sense of time may translate into increased stress and discomfort and affect recovery [10,16].

In the context of cancer, memories and plans are vital. Memories are a source of meaning and significance, providing comfort and hope in difficult moments. Reinterpreting memories allows for discovering new meanings, enhances inner strength, and enables more effective coping with difficulties. Planning for the future is a motivating force, allowing patients to maintain hope and purpose in the face of illness, providing a sense of control and a focus on values and life goals. At the same time, people struggling with chronic illnesses often feel confused and discouraged as their life plans and objectives have to be adjusted to their new health reality. This can lead to losing control and hope of achieving their goals. Reflecting on the human condition in the face of illness allows one to search for the meaning of life, consider suffering, death, and transcendence, and ultimately help one find answers to questions of identity and values in the face of illness [5,12,13,17,18,19,20,21,22,23].

Memories and plans for the future can be a source of strength and motivation to fight. It can be helpful for patients to focus on positive memories, such as moments spent with loved ones, life accomplishments, and goals they still want to achieve. Planning for the future, even in the face of the disease, can give patients a sense of control and hope, which are crucial in the struggle against cancer. On the other hand, patients may experience difficult memories of medical procedures, loss of health, and the unpleasant emotions accompanying treatment. They often feel the loss of their previous identity when aspects of their lives are altered or reduced [3,4,5,11,24,25,26,27,28,29,30,31].

### 1.2. Lived Time in Ovarian Cancer: Qualitative Phenomenology

Our (MM and JWB) previous qualitative study explored the lived experience of time among ovarian cancer patients while in chemotherapy [13,32]. The research was based on Giorgi’s method of phenomenological psychology and, among others, found two experiential structures affecting the lived experience of time, which we called “broken horizon” and “happiness closed off by regret”.

A broken horizon is a sense of general uncertainty in life, which leads patients to focus on the narrow present and short-term goals. A broken horizon means that those affected by ovarian cancer do not plan very far into the future. Their perspective covers only a few days, and their dreams of recovery are not precisely defined. As one patient put it (P9): *Unpredictable life has taught me the humility not to look too far ahead* [13].

The structure of regret-closed happiness was described as a combination of regret, guilt, and nostalgia for a lost, happy past, which now seems distant and inaccessible. It is the time before diagnosis associated with better days, good health, and disease-free life, and it becomes integral to present life during illness and includes an element of longing for the happy past [13,32]. As one patient put it (P7): *I wish I could live the way I lived when I didn’t have this problem(...) I had such a happy life* [13]. A sense of regret, on the other hand, comes from reflecting on what was not realized before the disease and what will no longer be possible.

The aim of this study was to investigate whether these two experiential phenomena regarding temporal experience are observed in a more extensive and oncologically more diverse sample. We were interested in finding whether they occur in the lived experience of other types of cancer, also because they may be relevant to fighting the disease, maintaining hope, and coping more effectively with difficulties.

## 2. Materials and Methods

### 2.1. Objectives

This is the second part of a sequential (first qualitative, then quantitative) study, whose aim is to explore the lived experience of past and future during oncological treatment. Following our previous qualitative results, we hypothesized that individuals undergoing chemotherapy will have difficulty in planning for the future and accurately defining their life goals and that they will often return in thought to their life before treatment with a sense of grief over the loss as well as in the form of happy memories.

### 2.2. Objective Temporal Variables

In addition, we were interested in seeing if there was a relationship between how time is subjectively experienced during chemotherapy and objective temporal variables in the form of time elapsed since diagnosis and the beginning of treatment (objectified linear time) and the frequency of chemotherapy sessions (objectified circular time). This was because, except for our previous studies, there is no research on the effect of chemotherapy length and treatment rhythm on cancer patients’ felt sense of time. In contrast, our previous studies showed that the treatment rhythm (rather than linear time) is critical to the experience of time [10,16,30].

### 2.3. Research Questionnaire

This study used a proprietary questionnaire that was designed to explore the experience of time in people being treated for cancer in accordance with hypotheses stemming from our previous qualitative study in a sequential manner (for the original version of the questionnaire, see the Appendix A). The questionnaire consisted of two parts. The first metrical part was designed to collect sociodemographic data and information directly related to the disease, such as the type of cancer, time passed since diagnosis and the beginning of chemotherapy, and the current frequency of chemotherapy courses. This part of the questionnaire made it possible to obtain context information for independent variables. The second part of the questionnaire contained seven statements about the reflective sense concerning the lived experience of time in the context of revisiting memories and planning for the future. These statements were directly related (as face-validated by MM and JWB) to the qualitative experiential structures from our previous study described in the introduction and called “broken horizon” and “happiness closed off by regret” [13]. Respondents were asked to rate the degree to which they agreed with each statement using a 7-point Likert scale. These statements were (translated here from Polish to English—see the Appendix A for the original phrasing): “I don’t think about the past because I focus on the present”; “Thoughts about my life before treatment often come back to me in the form of regrets. I regret that I did/did not do something that could have had negative consequences”; “Regardless of the above, thoughts of my life before treatment come back to me as happy memories”; “I live from day to day”; “I am not planning the future further than my next chemo”; “I have no concrete plans for the future after recovery, at most dreams”; and “I live with a constant sense that something unforeseen might happen.” In addition, in the same section of the questionnaire, the study participants were asked to define their horizon of the future in calendar terms (days, weeks, months, years) and how they measured time during treatment. The latter question was phrased: “During treatment, I measure time using,” with the options of calendar values (such as weeks or months) as well as treatment-related values, such as chemotherapy courses or follow-up visits, for contrast. These nominal-scale questions were designed to add to the understanding of the respondents’ lived experience of the future qualitatively explored in the previous study [13] by quantifying it in terms of calendar variables.

### 2.4. Data Collection

Data were obtained by engaging Polish-speaking patient groups from the social networking site Facebook, which were chosen based on their size and ongoing activity to maximize the response rate. The groups in question were as follows: Healed from cancer true stories; Beat cancer; Leukemia, lymphomas and other blood cancers support group; Patients and their loved ones—hematology department; Cancer is not a sentence. Join us if you have chosen to live for real; Cancer; Breast cancer—tame the fear. Support group for amazons; and Lung cancer. The Bioethics Committee of the Poznan University of Medical Sciences approved the study design as non-experimental. The entire study was conducted by the principles of the Declaration of Helsinki, and all participants gave their informed consent to participate.

### 2.5. Data Analysis

The values of the variables representing temporal experience phenomena on the ordinal scale were analyzed statistically and presented using the mean, median, and standard deviation, while the nominal-scale variables were presented using the count and percentage. In addition, quartiles were used to illustrate the characteristics of the group. A one-way analysis of variance (ANOVA) was performed to examine potential differences in the respondents’ questionnaire responses according to gender, age, place of residence, education, and type of cancer. The normality of the distribution was assessed using the W Shapiro–Wilk test. The Mann–Whitney U test was used to compare independent groups, Spearman’s R correlation was used to assess the relationship between linear time variables and ordinal scale temporal phenomena, and the chi-square test was used to compare groups with different cancers and different treatment frequencies and their temporal experience represented by the nominal-scale variables. A significance level of *p* < 0.05 was adopted for statistically relevant differences or relationships. Database and statistical tests were conducted using the STATISTICA 13.0 computer software (StatSoft, Kraków, Poland).

## 3. Results

### 3.1. Sample Characteristics

A group of 202 patients with a cancer diagnosis participated in this study. Women predominated among the respondents, accounting for 87.13% of the total group (n = 176), while men accounted for 12.87% (n = 26). A total of 15.84% (n = 32) of the participants were under 30 years old, 13.37% (n = 27) were between 31 and 40 years old, 31.68% (n = 64) were between 41 and 50 years old, 27.72% (n = 56) were between 51 and 60 years old, and only 11.39% (n = 23) were patients over 60 years old. The majority of the respondents had a high school (24.75%) or college education (41.09%). An overwhelming number of the participants lived in cities with a population of up to 500,000, accounting for 20.79% of the study group. In terms of family income, the majority of the respondents had an income of up to PLN 3000 (34.65%) or up to PLN 5000 (30.69%). In terms of the type of cancer, the largest number of respondents suffered from breast cancer (22.27%). High incidence rates were also recorded for ovarian cancer (10.89%) and cervical cancer (10.40%). In contrast, single cases included brain tumors (0.50%), anaplastic thyroid cancer (0.50%), nasopharyngeal cancer (0.50%), rectal cancer (0.50%), bladder cancer (0.50%), penile cancer (0.50%), and salivary gland cancer (0.50%)—see Table 1.

The most common chemotherapy administration regimen was a triweekly (31.68%) or biweekly (28.22%) cycle, with 16.83% of the patients receiving chemotherapy once a week and 3.96% receiving it daily.

The median time since diagnosis was 18 months (Me 18), with a mean of just over two years, while the time range varied significantly, indicating the heterogeneity of the respondents (M 25.63, SD 28.88). The median time from the beginning of chemotherapy was 7.5 months (Me 7.5), with a mean of about a year (M 12.53, SD 12.44). These differences reflect the diversity of the patients’ experiences of time in terms of their duration (in the linear sense of time), regardless of the content.

### 3.2. Happiness Closed off by Regret and Broken Future Horizon

Based on the assumption that the 7-point Likert scale was sensitive enough to represent the subtle differences among the respondents, even if the final numerical values merely concerned the level of agreement with the pre-given content (and were, therefore, not absolute), we looked at numbers above the median values of the 7-point Likert (i.e., representing neither agree nor disagree) as indicating an overall tendency to have the experience in question (as represented by each particular variable statement). A slight tendency to accept the statements representing the phenomena of happiness closed off by regret and broken future horizon was apparent, with all the responses’ median values being higher than the means (see Table 2).

The median score of 5.0 regarding focusing on the present indicates that most participants experienced their lives during treatment in a continuous, day-to-day manner, which was also confirmed by the agreement with the statement on living from day to day, with a median score of 5.0. This may, however, suggest the patients’ ability to enjoy the present moment despite difficulties. Reflections on life before treatment in the form of regrets were less intense, with a median score of 4.0. In contrast, the statement regarding the thoughts of life before treatment coming back as happy memories received a median score of 5.0. With regard to planning for the future, the median values were lower, with 4.0 for not planning further than the next chemotherapy treatment and 5.0 for not having any specific plans after recovery. The median score regarding living with a constant sense that something unforeseen might happen was 5.0, suggesting that half of the respondents experienced uncertainty related to the future (see Table 2).

The participants’ future horizon specified in calendar terms was relatively short. A total of 23% of the respondents looked ahead no further than six months, while 20.79% looked ahead no further than the coming week. For 14.84%, the horizon reached the next three months; for 12%, the upcoming chemotherapy; and for 9%, the next day only. Only one out of five respondents (19.31%) reached further than a year, including longer periods.

The dominant method of measuring time during treatment could not be identified in the study group. Nevertheless, the most common was the rhythm of check-ups (29%), the monthly cycle (26%), the schedule of chemotherapy courses (17%), the daily rhythm (15%), and the weekly rhythm (13%). It is noteworthy, however, that measuring time by non-calendar rhythms related to treatment, that is, chemotherapy and check-ups, applied to almost half of the respondents (46%), thus supporting our previous hypothesis [32].

Interestingly, no statistically significant differences were observed between the cancer groups and the responses to any of the questions on experienced temporal phenomena (*p* > 0.05). What is more, no sociodemographic variables showed statistically significant differences (*p* > 0.05) with the temporal variables representing the lived experience of time. In this context, it can be concluded that neither the type of cancer nor the social background was a significant factor in differentiating the temporal experience of the patients.

### 3.3. Correlations with Cyclic and Linear Time

The frequency with which the patients underwent chemotherapy only affected their focus on the present, and mildly (Being in the present; R = 0.25, *p* < 0.05), likely because of the discomfort of the side effects. The frequency of chemotherapy also did not differentiate the results of the responses to the question concerning how time is measured—which was our original hypothesis, confirmed in another study [16]—and the question concerning the length of the temporal horizon in calendar terms (Chi2, *p* > 0.05).

As far as linear time is concerned, there were significant but very weak relationships between age and three variables: Bygone happiness (R = 0.18, *p* < 0.05), Future planning (R = 0.16, *p* < 0.05), and Unpredictable future (R = 0.14, *p* < 0.05). No significant relationship was observed between age and the other statements (*p* > 0.05). Except for Dreams about the future (R = 0.17, *p* < 0.05), there were no correlations between the time passed since diagnosis or the beginning of chemotherapy and the rating of the statements (see Table 3).

However, all of these correlations were negligible, which can be interpreted to mean that entering the cancer treatment stage reevaluates the attitude toward time regardless of the type of cancer, the age of the patient, and the length of the process in calendar time. In this sense, chemotherapy affects lived temporality totally and equally for everyone.

## 4. Discussion

At the outset, it should be noted that chemotherapy treatment is a traumatic experience for any patient, negatively affecting their mental state. The patient usually dramatically changes their attitude to their surrounding reality and social relations. This study aimed to explore, in a sequential manner, following our previous qualitative phenomenological study concerning ovarian cancer, how patients in a larger and more diverse sample experience time during chemotherapy in terms of their past memories and future plans.

The analysis showed that for almost half of the respondents, the rhythm of chemotherapy courses and check-ups was the means of measuring the passing of time—what we previously called the chemo-clock [32]. This both indicates the subjective significance of treatment for the patients and potential adverse psychological effects. Frequent and meaningful hospital visits might lead to a sense of security and optimism and, in turn, positively influence the course of treatment and its outcomes. Still, they may also make patients more susceptible to worries and anxieties about aspects of treatment, such as the proper intake of medication, the body’s readiness for subsequent doses, and concerns about the possible return of the disease.

The frequency of chemotherapy only mildly affected the respondents’ perception of the present, with more frequent cycles likely leading to a focus on current aspects of the disease. But, even with frequent cycles, no significant differences were observed regarding other aspects of the lived experience of time. Nevertheless, a slight tendency to agree with all the questions points to how patients’ temporal experience changes during treatment. The increased present focus might be a defensive element, in that by focusing on the current stage of treatment, they isolate themselves from negative memories of the past and an uncertain future. Focusing on the present is a way to cope with these emotional challenges [33]. Simultaneously, the respondents’ narrowed future planning likely stems from the need to face treatment challenges and future uncertainties. Immediate goals and plans within their reach are what counts. It is worth noting that even slight differences in the tendency to think ahead can have a significant impact on the psychological state of patients. A sense of hope and control can be an important element in coping with the difficulties arising from the disease, affecting their ability to adapt to a new situation. A limited and unpredictable future horizon may lead to avoiding activities related to uncertainties about the disease and its treatment. Directing attention to the present may be due to physical ailments. Still, it is a way of coping with difficulties while also effectively counteracting negative emotions related to an uncertain future. In addition, research highlights the contrast between revisiting happy memories, where the past appears as a separate reality, and the present, where feelings of anxiety, apprehension, and uncertainty dominate. This may induce patients to cling to dreams instead of concrete plans, as uncertainty about the course of life complicates the formulation of realistic expectations. An interesting aspect is the observation of feelings of regret resulting from reflections on decisions made before the onset of the disease, decisions that could have potentially brought negative consequences. This phenomenon can be interpreted as a sense of deserved punishment for past behavior or a belief that there was a conscious cause for the onset of the disease.

This study identified some differences in temporal experience related to the type of cancer; however, they were not statistically significant. Respondents with blood and squamous cell cancers were more likely to agree with the statements regarding past happiness and living for today, and their temporal horizon was more limited by the upcoming chemotherapy. This is likely related to the rapid growth and expansion of these cancers, which requires a decisive therapeutic approach within a short period after diagnosis. Also, the long-term and chronic treatment of these types of cancer requires patients to be constantly monitored and controlled, potentially affecting their outlook on time. Focusing on the present may effectively cope with difficulties and counteract negative emotions about the future. The diagnosis of these types of cancer is often made at an advanced stage, which increases patients’ awareness of the limitations and future difficulties. Reminiscence may be a way to cope with the disease and its consequences. It may provide a kind of escape from the anxiety associated with the possibility of recurrence. The aspect of past happiness is also relevant in the context of breast cancer, where anticancer therapies often require involvement in very invasive procedures, which can lead to visible effects on the body. Further studies in larger and more diverse samples are needed to confirm these hypotheses. Currently, the type of cancer does not appear to be a significant factor affecting the temporal experience during treatment.

In terms of linear time, elderly cancer patients experience more frequent horizon collapses. Lack of hope for recovery, worries about the future, and probably awareness of imminent death, as well as more frequent revisiting of the past, are more common among the elderly, although correlations are low. Arguably, cancer is a factor that usually compels the patient to reflect on the past and related memories, both positive and difficult. The changed life situation places high demands on the ability to adapt and cope with daily life and redefine one’s social roles. These changes involve a sense of existential change and looking at oneself differently. People also experience thoughts related to dying and death. More time is spent reflecting on life to date, living with cancer, and death [4,34,35,36,37]. The younger age of patients, on the other hand, mildly translates into a broader time perspective. Cancer is usually seen as a challenge to overcome or to live with. Young patients actively analyze the possibility of relapse and its potential impact on their family life, finances, image, sexual function, and future roles in the family. Faced with these uncertainties, they seek to flexibly shape their experiences, adjusting priorities in their professional and family lives. Planning for the future becomes a motivating force for them, enabling them to maintain hope, a sense of control, and focus on their values and life goals, which promotes effective coping [37,38,39,40,41,42,43,44]. Furthermore, in terms of the linear time passed since diagnosis and treatment, there are only subtle connections between the former and dreaming about the future. This may result from less hope for recovery caused by deterioration, chronicity of the disease, or related limitations. It is noteworthy that despite fears and anxieties about taking further doses of drugs or concerns about the body’s readiness for therapy, the duration of chemotherapy did not affect how time was experienced.

### Limitations

The main limitation of this study is the sample size and convenience sampling method, which is never representative. Another limitation is that the participants consisted of online cancer support groups only, with a possible overrepresentation of people with online access. Furthermore, there was a female tumor prevalence, namely breast, ovarian, and cervix. For these reasons, and also given that the questionnaire was constructed for the purpose of this study only in order to account for hypotheses stemming from our previous small-sample qualitative research, the results of this follow-up quantitative study should be treated as exploratory. Further studies in larger and more diverse cancer samples, and possibly directly collected data, are needed to confirm these results.

## 5. Conclusions

During chemotherapy, patients experience increased focus on the present, decreased focus on the future, and a sense of unpredictability with a relatively short temporal horizon. They also tend to measure time during treatment by the rhythm of chemotherapy and check-ups. The type of cancer and the length of treatment do not appear to be a significant factor affecting the temporal experience of patients. Changed temporal experience during chemotherapy should be considered when planning oncology care.

## Figures and Tables

**Table 1 cancers-16-02124-t001:** Sample characteristics (n = 202).

		N (%)
Education level		
	Basic Education	11 (5.44)
	Vocational Education	33 (16.33)
	Secondary Education	50 (24.75)
	Higher Education	83 (41.09)
	PhD	18 (8.91)
	Higher Medical Education	7 (3.46)
Place of residence		
	>500 k	25 (12.38)
	<500 k	42 (20.79)
	<250 k	40 (19.80)
	<100 k	37 (18.32)
	<20 k	37 (18.32)
	<5 k	21 (10.39)
Age		
	<30	32 (15.84)
	31–40	27 (13.37)
	41–50	64 (31.68)
	51–60	56 (27.72)
	>60	23 (11.39)
Sex		
	Female	176 (87.13)
	Male	26 (12.87)
Chemotherapy frequency		
	4 weeks or more	25 (12.38)
	3 weeks	64 (31.68)
	2 weeks	57 (28.22)
	1 week	34 (16.83)
	Everyday	8 (3.96)
	Last	9 (4.46)
	Interrupted	2 (0.99)
	Starting	3 (1.49)
Type of cancer		
	Breast	45 (22.27)
	Ovarian	22 (10.89)
	Cervical	21 (10.40)
	Colorectal	19 (9.41)
	Leukemia	17 (8.41)
	Lung	12 (5.94)
	Kindey	11 (5.45)
	Hodgkin’s	11 (5.45)
	Other (Testicular, Lumphotic, Liver, Multiple Myeloma, Prostate, Uterus, Melanoma, Pancreatic, Skin, In situ, AML, Penile, Salivary gland, Bladder, Nasopharyngeal, Anaplastic thyroid, Brain)	44 (21.78)

**Table 2 cancers-16-02124-t002:** Lived time phenomena and their representing statements.

Phenomenon	Statement	Value M, Me, SD, Q1, Q2, Q3
Being in the present	I don’t think about the past because I focus on the present.	4.11, 5.00, 1.19, 3.00, 5.00, 5.00
Regret	Thoughts about my life before treatment often come back to me in the form of regrets. I regret that I did/did not do something that could have had negative consequences.	3.55, 4.00, 1.47, 2.00, 4.00, 5.00
Bygone happiness	Regardless of the above, thoughts of my life before treatment come back to me as happy memories.	4.29, 5.00, 1.00, 3.00, 5.00, 5.00
Living for today	I live from day to day.	4.05, 5.00, 1.37, 3.00, 5.00, 5.00
Future planning	I am not planning the future further than my next chemo.	3.51, 4.00, 1.58, 2.00, 4.00, 5.00
Dreams about future	I have no concrete plans for the future after recovery, at most dreams	3.99, 5.00, 1.36, 3.00, 5.00, 5.00
Unpredictable future	I live with a constant sense that something unforeseen might happen	4.14, 5.00, 1.18, 3.00, 5.00, 5.00

**Table 3 cancers-16-02124-t003:** Spearman’s R correlations between linear time variables and temporal phenomena.

Linear Time Variables	Being in the Present	Regret	Bygone Happiness	Living for Today	Future Planning	Dreams about Future	Unpredictable Future
Age	0.11	0.12	**0.18 ***	0.10	**0.16 ***	0.04	**0.14 ***
Time since diagnosis	0.01	0.00	−0.04	−0.03	0.09	**0.17 ***	0.00
Time since the beginning of chemotherapy	0.01	−0.02	0.01	−0.08	−0.11	0.01	−0.4

Bold * indicates statistically significant results (*p* < 0.05).

## Data Availability

Data available upon request.

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
