# Peer review of "Past Happiness and Broken Future Horizon of Oncological Patients during Chemotherapy—A Quantitative Exploration of a Phenomenological Hypothesis"

_cancers, 2024, doi:10.3390/cancers16112124_

Round 1
Reviewer 1 Report
Comments and Suggestions for Authors
Interesting quantitative study but could be improved in the following points:
Although the sample is small, it would be interesting if the authors applied basic inferential statistics to indicate whether there were any significant differences by gender, age groups or even type of neoplasm.
Table 1 should indicate the meaning of the acronyms at the foot of the table.
The questionnaires applied should be included in the metadata and in the material and methods they should adequately justify why they selected these questionnaires and not others.
In general, it can be published with these corrections.
Author Response
Many thanks for the useful comments that helped us improve the paper. A direct reply to your particular points below:
- “Although the sample is small, it would be interesting if the authors applied basic inferential statistics to indicate whether there were any significant differences by gender, age groups or even type of neoplasm”
Reply: We conducted a one-way analysis of variance to examine respondents' answers to the questionnaire, taking into account differences in gender, age, education, and type of cancer. The analysis showed no significant differences in the answers given. The results indicate that regardless of gender, age, education level, and type of cancer, respondents responded similarly, and we have included this information in the paper.
- “Table 1 should indicate the meaning of the acronyms at the foot of the table.”
Reply: As suggested, the meanings of acronyms in the footer of Table 1 have been added, which improves readability and understanding of the content. In addition, definitions of phenomena have been included in Table 2, which further facilitates understanding of the contents of Table 3.
- “ The questionnaires applied should be included in the metadata and in the material and methods they should adequately justify why they selected these questionnaires and not others.
Reply: We have added an additional explanation of the methods and included the original version of the questionnaire in the supplementary files while explaining the questions in more detail in English in the manuscript.
Reviewer 2 Report
Comments and Suggestions for Authors
In this article, the authors delve into the psychological well-being of cancer patients, a topic of utmost importance and relevance. This discussion is crucial for enhancing the support that can be extended to patients.
This work raises several significant issues that the authors could reconsider or modify. I suggest a major revision to provide more specific feedback.
Materials and methods section:
The objective of the study and the method applied need to be better defined.
Paragraph 2.2 does not seem consistent with section 2. Perhaps it could be moved to the introduction to better "introduce" the reader to the methodology and the expected results.
Paragraph 2.4: The authors could better describe the characteristics of the questionnaire (or survey?). Some hints: Is it a questionnaire created by the authors? Is it a validated instrument? How are the Likert scale scores and the resulting score interpreted?
Section 2.5. In this paragraph, the major criticisms are gathered partly from the limitations stated by the authors. However, as they consider the work to be an exploratory study, I suggest implementing the description of the data collection by describing how the Facebook groups were chosen, what type of "topic" was chosen, and how this choice influenced a specific type of patient. The patients included, which centre did they refer to, and were the care facilities similar?
Section 2.6. "the value of the measurable parameters" What parameters? What information was collected?
Results section
The selection of tumour types , mainly of female prevalence (breast, ovary and cervix), influenced the percentage of respondents represented by women. Furthermore, the choice to include subjects who use social media may have influenced the selection of younger subjects or those similar to "technological" devices; only 23 patients were over 50.
In lines 198-200, the description of time diagnosis could be clearer. I would reformulate by placing the standard deviation alongside the mean and reporting the data relating to the median separately. However, why do the authors indicate both measures of central tendency? A mean of 25.63 and a median of 18 are reported, suggesting an asymmetry in the data's distribution. So why not just use the median?
Table 1: The percentages indicated must all be checked. At the education level variable, the total expected percentage of 100% is 114%. Something may have gone wrong in the BC category. Similarly, I found a typo in the percentage of the "Over 60" category in variable age and Sex "M".
For the missing data variables, consider adding the data to the table or indicating in notes which contains them and how many.
The full definition of the education level categories could also be added to the notes for clarity.
Finally, for consistency, consistently report the same number of decimal places.
Paragraph 3.2 This paragraph would be more straightforward if the interpretation of the score had been better explained in the methods section.
Table 2 The mean is deprived of its standard deviation. Why was the IQR chosen instead of the range (I-III quartile) or min max for the median? Were roof or floor effects observed in the questionnaire responses?
How many subjects responded to the questionnaire? Table 1 shows that the type of tumour is unknown for 22% of the patients included in the study (missing data). Did these subjects answer the questionnaires?
Author Response
Many thanks for the useful comments that helped us improve the paper. A direct reply to your particular points below:
-“The objective of the study and the method applied need to be better defined”.
Reply: We have significantly expanded the idea behind the study and the methods section.
- “Paragraph 2.2 does not seem consistent with section 2. Perhaps it could be moved to the introduction to better "introduce" the reader to the methodology and the expected results”
Reply: Yes, this is a good idea, the paragraph has been moved and expanded.
-“Paragraph 2.4: The authors could better describe the characteristics of the questionnaire (or survey?). Some hints: Is it a questionnaire created by the authors? Is it a validated instrument? How are the Likert scale scores and the resulting score interpreted?”
Reply: We have added additional information regarding the questionnaire and included the original version of the questionnaire in the supplementary files while explaining the questions in more detail in English in the manuscript. The choice of a 7-point Likert scale was dictated by its proven effectiveness and high sensitivity to subtle differences in respondents' attitudes. Our goal was for the questionnaire to accurately measure the degree of respondents' agreement with particular statements, thus providing a deeper understanding of those aspects of their experience of time that stemmed from our previous qualitative research (as now explained in more detail as well). We also added a commentary on interpreting the Likert scale scores in the results.
-“Section 2.5. In this paragraph, the major criticisms are gathered partly from the limitations stated by the authors. However, as they consider the work to be an exploratory study, I suggest implementing the description of the data collection by describing how the Facebook groups were chosen, what type of "topic" was chosen, and how this choice influenced a specific type of patient. The patients included, which centre did they refer to, and were the care facilities similar?”
Reply: We have included more information in the data collection part as well as expanded the limitations section to account for this justified remark.
-“Section 2.6. "the value of the measurable parameters" What parameters? What information was collected?
Reply: We have corrected this section to give a better account of the types of variables (nominal, ordinal) used in the survey, and which variables were treated as independent.
- “The selection of tumour types , mainly of female prevalence (breast, ovary and cervix), influenced the percentage of respondents represented by women. Furthermore, the choice to include subjects who use social media may have influenced the selection of younger subjects or those similar to "technological" devices; only 23 patients were over 50.”
Reply: Thank you for your comments. However, actually, there were 23 patients over 60 (and not 50). We agree that technology could slightly influence the selection of younger people to participate in the survey, taking into account the more frequent use by younger people. Nevertheless, the group of people aged 41-50 is the most numerous (64 respondents), and those aged 51-60 are also a significant part (56 respondents). A look at Table 1 shows that respondents, even if not directly age-representative, actually vary across ages. In terms of the diversity of cancer types, we agree that the majority of respondents suffer from breast cancer. Nevertheless, other types, such as lung cancer, leukemia, colorectal cancer, and Hodgkin's disease, are also present in the remaining survey participants. This variation adds to the value of the sample, but of course, a convenience sample is never representative. We have added an additional explanation of this issue in the limitation section.
-“In lines 198-200, the description of time diagnosis could be clearer. I would reformulate by placing the standard deviation alongside the mean and reporting the data relating to the median separately. However, why do the authors indicate both measures of central tendency? A mean of 25.63 and a median of 18 are reported, suggesting an asymmetry in the data's distribution. So why not just use the median?”
Reply: In accordance with your suggestion, we have made the appropriate adjustments. We believed that the presentation of both measures of central tendency would better show the heterogeneity of the group and the varied experiences of patients, which in turn affected respondents' perceptions of time
-“Table 1: The percentages indicated must all be checked. At the education level variable, the total expected percentage of 100% is 114%. Something may have gone wrong in the BC category. Similarly, I found a typo in the percentage of the "Over 60" category in variable age and Sex "M". For the missing data variables, consider adding the data to the table or indicating in notes which contains them and how many.
-The full definition of the education level categories could also be added to the notes for clarity.
-Finally, for consistency, consistently report the same number of decimal places. “
Reply: In view of your comments, we made the appropriate corrections. We confirm the completeness of the data with no gaps. The table was revised to maintain consistency in data presentation. We have ensured consistent formatting of the number of decimal places. In addition, we added explanations of acronyms in the footer of the table to make it easier for readers to understand the content.
-Paragraph 3.2 This paragraph would be more straightforward if the interpretation of the score had been better explained in the methods section.
Reply: In accordance with your suggestion, we have expanded the interpretation of the measures described in this paragraph and more fully described the statistical procedures used in the methods section.
-“Table 2 The mean is deprived of its standard deviation. Why was the IQR chosen instead of the range (I-III quartile) or min max for the median? Were roof or floor effects observed in the questionnaire responses?”
Reply: Following your suggestion, we added quartiles and standard deviations to better understand the distribution of the data. Accordingly, Table 2 has been updated.
-“ How many subjects responded to the questionnaire? Table 1 shows that the type of tumour is unknown for 22% of the patients included in the study (missing data). Did these subjects answer the questionnaires? “
Reply: There were 202 respondents to our survey. 22% of the cases involved types of cancer that appeared in only one, two, or sometimes three respondents – hence, we created a category “other.” We have now given the details in Table 1.